# Regional heterogeneities of oligodendrocytes underlie biased Ranvier node spacing along single axons in sound localization circuit

**Ryo Egawa[1], Kota Hiraga[1], Ryosuke Matsui[2], Dai Watanabe[2], Hiroshi Kuba[1]***

[1]Department of Cell Physiology, Graduate School of Medicine, Nagoya University, Nagoya, Japan; [2]Department of Biological Sciences, Graduate School of Medicine, Kyoto University, Sakyo-ku, Kyoto, Japan

*For correspondence:
kuba@med.nagoya-u.ac.jp

Competing interest: The authors declare that no competing interests exist.

## eLife Assessment

This **important** study uses the delay line axon model in the chick brainstem auditory circuit to examine the interactions between oligodendrocytes and axons in the formation of internodal distances. This is a significant and actively studied topic, and the authors have used this preparation to support the hypothesis that regional heterogeneity in oligodendrocytes underlies the observed variation in internodal length. In a **solid** series of experiments, the authors have used enhanced tetanus neurotoxin light chains, a genetically encoded silencing tool, to inhibit vesicular release from axons and support the hypothesis that regional heterogeneity among oligodendrocytes may underlie the biased nodal spacing pattern in the sound localization circuit.
[Editors' note: this paper was reviewed by Review Commons.]

**Abstract** Spacing of Ranvier nodes along myelinated axons is a critical determinant of conduction velocity, influencing spike arrival timing and hence neural circuit function. In the chick brainstem auditory circuit, the pattern of nodal spacing varies regionally along single axons, enabling precise binaural integration for sound localization. Using this model, we investigated the potential factors underlying the biased nodal spacing pattern. 3D morphometry revealed that these axons were almost fully myelinated by oligodendrocytes exhibiting distinct morphologies and cell densities across regions after hearing onset. The structure of axons did not affect internodal length. Inhibiting vesicular release from the axons did not affect internodal length or oligodendrocyte morphology, but caused unmyelinated segments on the axons by suppressing oligodendrogenesis near the presynaptic terminals. These results suggest that the regional heterogeneity in the intrinsic properties of oligodendrocytes is a prominent determinant of the biased nodal spacing pattern in the sound localization circuit, while activity-dependent signaling supports the pattern by ensuring adequate oligodendrocyte density. Our findings highlight the importance of oligodendrocyte heterogeneity in fine-tuning neural circuit function.

## Introduction

Ranvier nodes are highly excitable domains distributed along myelinated axons and contribute to increased conduction velocity through saltatory conduction. The distance between adjacent nodes (internodal length) affects the speed of saltatory conduction, with longer internodes leading to faster conduction, and hence has a significant impact on neural circuit function by influencing the timing of

input arrival at target neurons. The pattern of nodal spacing along axons is not necessarily uniform. Recent studies have shown that the internodal length varies not only among axonal types and brain regions (*Chong et al., 2012*), but also along individual axons (*Tomassy et al., 2014*; *Ford et al., 2015*; *Bonetto et al., 2021*). Despite its potential importance in neural information processing, the mechanisms regulating nodal spacing patterns are not well understood (*Normand and Rasband, 2015*).

Various factors could contribute to the emergence of nodal spacing patterns along axons. In the central nervous system (CNS), each Ranvier node is formed through the following processes. Immature oligodendrocytes contact axons with their processes, initiating myelin formation that extends along the axon (*Snaidero et al., 2014*). At the ends of the extending myelin, the paranodal domain, at which the myelin tip contacts the axon, acts as a diffusion barrier and facilitates the clustering of voltage-gated sodium channels (Nav) and the scaffolding protein ankyrinG (AnkG) to form a heminode. Subsequent restriction of the gap between two neighboring heminodes forms a mature Ranvier node (*Vabnick et al., 1996*; *Susuki et al., 2013*; *Rasband and Peles, 2015*; *Rasband and Peles, 2021*). Thus, the patterning of nodal spacing along axons is based on multicellular interactions among oligodendrocytes and axons and could be influenced by multiple factors associated with these cells. The factors related to oligodendrocytes include their morphological characteristics (e.g., the number and length of myelin sheaths), and cell density relative to axons (*Chong et al., 2012*), while those related to axons include their branching (*Stedehouder et al., 2019*), diameter (*Bechler et al., 2015*), and neural activity (*Fields, 2015*; *Bechler et al., 2018*; *Bonetto et al., 2021*; *Osanai et al., 2022*). In addition, the contributions of these factors would vary significantly due to the diversity of both cell types and developmental stages, which obscures our understanding of the mechanisms of determining the nodal spacing patterns.

To understand the regulatory mechanisms of nodal spacing, the chicken brainstem auditory circuit is an excellent model because the nodal spacing is biased in a region-dependent manner along the same projecting axons, minimizing the effects of neuronal diversity. In addition, this regional difference in internodal length is also important in the computation of neural circuits for sound localization. Sound localization is the ability to identify the direction of sound source and involves detecting the time difference in sound arrivals between the ears (interaural time difference [ITD]) in the order of microseconds. The ITD detection is mediated by an array of coincidence detector neurons in the nucleus laminaris (NL) that receive excitatory synaptic inputs from both sides of the nucleus magnocellularis (NM), a homologue of mammalian anteroventral cochlear nucleus (*Figure 1A*; *Carr and Konishi, 1990*; *Hyson, 2005*). In this circuit, the spacing of nodes along the axon of NM neurons differs among regions; the internodal length is longer at the tract region across the midline than at other regions including the NL (*Figure 1B*; *Seidl et al., 2010*). The long internodes reduce the conduction time from the contralateral side and compensate for the difference in axonal pathlengths between the two sides (*Seidl et al., 2014*), while the short internodes expand the dynamic range for detectable ITD within the NL. Therefore, this model will be an important stepping stone for elucidating the regulatory mechanisms of nodal spacing and their impact on neural circuit function.

In this study, we tested the potential factors contributing to the regional differences in the nodal spacing along NM axons using high-resolution 3D morphometry of the optically cleared brainstem auditory circuit. The results showed that NM axons are almost fully myelinated by oligodendrocytes with distinct morphological features after hearing onset. This morphological difference was not due to axonal structure. Inhibition of vesicular release from NM axons did not affect the internodal length and oligodendrocyte morphology but caused unmyelinated segments on NM axons via a suppression of oligodendrogenesis at the NL region. These results identified that the major factor contributing to the biased nodal spacing pattern in the ITD circuit is the regional heterogeneity in the intrinsic properties of oligodendrocytes. Activity-dependent signaling also contributed to this circuit by ensuring the density of oligodendrocytes through enhanced oligodendrogenesis. Our findings provide new insights into the mechanism of regulating nodal spacing along single axons and also into the significance of oligodendrocyte heterogeneity in neural circuit function.

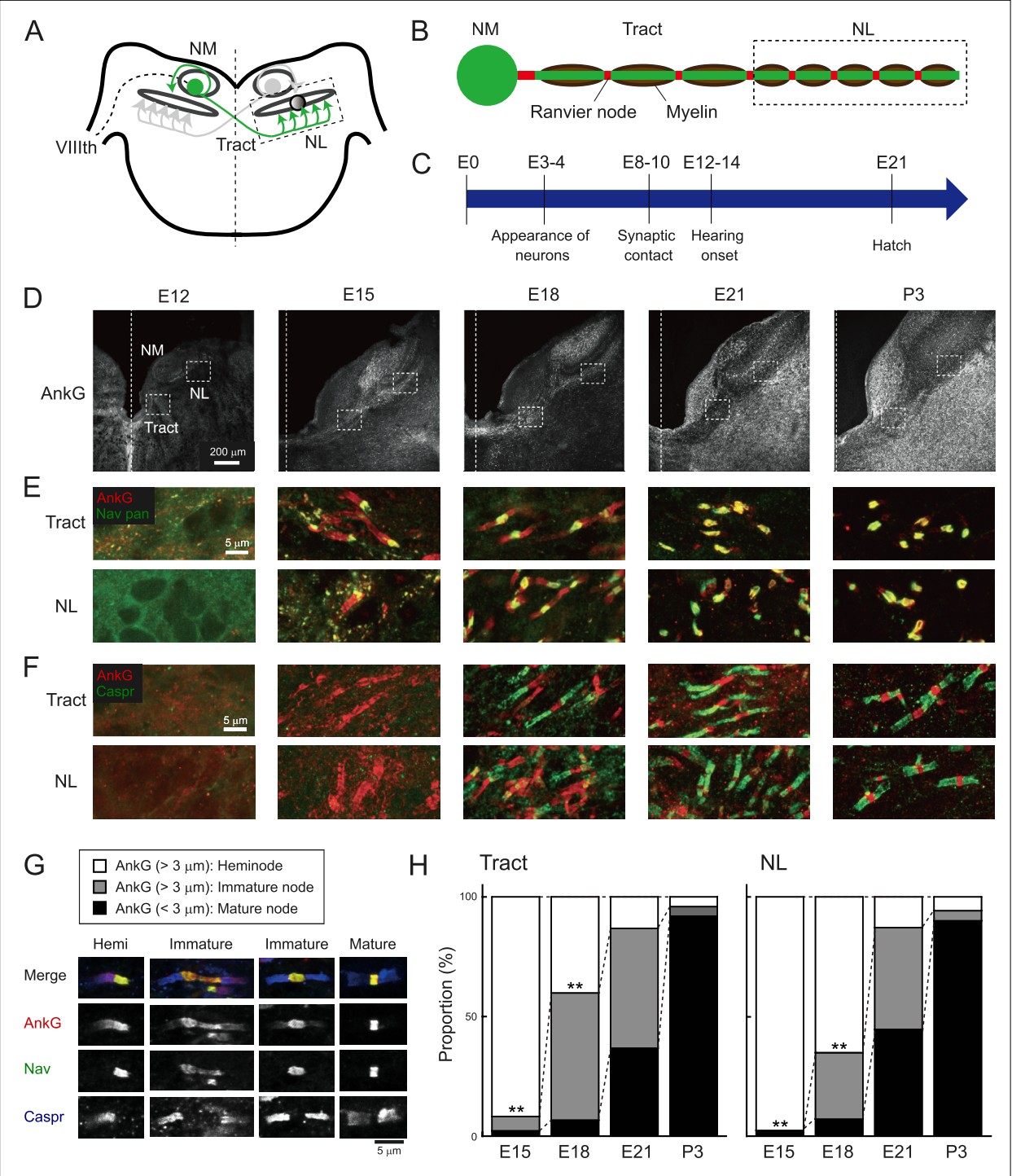

**Figure 1.** Development of Ranvier nodes progressed in a similar time course along nucleus magnocellularis (NM) axons after hearing onset. (**A**) Axonal projection of NM neurons in the chicken brainstem. NM axon projects to both sides of nucleus laminaris (NL) and forms 'delay line' at contralateral NL. VIIIth, auditory nerve. (**B**) Regional bias in periodic spacing of Ranvier nodes along the main trunk of NM axon. Internodal length is long at the midline tract region and becomes shorter at the NL region in the axon.(**C**) Development of NM neurons. NM neurons make synaptic contact with NL neurons by E10 and receive synaptic input from the auditory nerve by E12. (**D**) Immunostaining of AnkG between E12 and P3. Dashed line indicates midline. (**E, F**) Double immunostaining of AnkG (red) and panNav (green) for (**E**), and AnkG (red) and Caspr (green) for (**F**) at tract (upper) and NL (lower) regions from the same slices. (**G, H**) Ranvier nodes matured on a similar time course across the region. Three types of Ranvier nodes immunostained with AnkG (red), panNav (green), and Caspr (blue) antibodies (**G**) and their proportions between E15 and P3 (**H**). These types were determined according to the length of AnkG signals; the signals longer than 3 μm were defined as heminode or immature node according to the number of Nav-negative paranodal domains,

*Figure 1 continued on next page*

*Figure 1 continued*

and those shorter than 3 µm were defined as mature node. Note that some immature nodes had long nodal domains that exceeded 5 µm. This may correspond to a gap between the two heminodes. E15: n=183 and 177 nodes for tract and NL, N=4 chicks; E18: n=344 and 267 nodes for tract and NL, N=4 chicks; E21: n=267 and 371 nodes for tract and NL, N=4 chicks; P3: n=226 and 412 nodes for tract and NL, N=3 chicks. Scale bars: 200 µm (D) and 5 µm (E–G). Statistical analysis: chi-square test (**H**) was used to compare the proportions of node types between the regions at each developmental stage. **p<0.01.

The online version of this article includes the following source data for figure 1:

**Source data 1.** Quantitative measurements with associated statistical analyses underlying *Figure 1H*.

## Results

### Ranvier nodes developed in a similar time course along NM axons after hearing onset

NM neurons in chick embryos form axonal projections to the NL at embryonic day 10 (E10), begin to receive auditory input from around E12, and are functionally mature by the time around hatch (*Akter et al., 2018*; *Akter et al., 2020*; *Figure 1C*), allowing the animals to localize sounds immediately after hatching (*Grier et al., 1967*). To clarify how node formation progresses along development and whether there are regional differences in the timing of node formation, we examined the development of nodes at the tract and NL regions by double staining with AnkG and pan Nav antibodies or AnkG and Caspr antibodies, a marker for the paranodal domain (*Figure 1D–F*).

At E12, we did not find these molecules at each region. At E15, Nav-positive clusters appeared, and they were localized specifically at one end of the fibrous AnkG signals. The remaining Nav-negative part of AnkG signals would correspond to the paranodal domain, as myelin sheath at the domain shows transient expression of AnkG during the immature period of CNS (*Chang et al., 2014*), and these structures were defined as 'heminode'. At E18, Caspr signals became detectable at the paranodal domain of heminode. In addition, some Caspr signals appeared on both sides of the fibrous AnkG signals, sandwiching the Nav cluster. These structures would be formed through the fusion of adjacent heminodes and were defined as 'immature node'. At E21, many nodes showed mature patterns, with complete colocalization of Nav and AnkG at a short nodal domain, which was flanked by Caspr-positive paranodal domains ('mature node'). Triple staining of these molecules confirmed the three stages of nodal development along the axon of NM neurons. Quantification of their abundance revealed that their maturation progressed gradually after hearing onset and was almost completed around hatch. 90% of the node types were heminodes at E15. This decreased to about 50% by E18 and to about 10% by E21 (*Figure 1G and H*), suggesting that NM axons are almost fully myelinated by E21. Although the increase in the proportion of immature nodes was slightly preceded at the tract region, the overall tendency did not differ greatly between the regions. These data indicate that the formation of Ranvier nodes progresses in a similar time course along the axon.

### Development of myelin sheath along NM axons

Formation of Ranvier nodes requires maturation of myelin sheaths. To clarify their relationship on NM axons, we also examined the development of myelin with an antibody against myelin-associated glycoprotein (MAG) during the period of node formation.

The immunosignals for MAG first appeared around the midline of the brainstem at E12 and extended along the axons of NM neurons by E15, becoming more intense at later ages for both tract and NL regions (*Figure 2A and B*). Detectable AnkG-positive clusters along NM axons also emerged at E13 concurrent with MAG signals, albeit very sparsely (*Figure 2—figure supplement 1*). These observations are consistent with the time course of nodal development shown in *Figure 1* and with the concept that Ranvier nodes form as myelin sheaths mature. Importantly, many oligodendrocytes were situated along the axon, as identified by immunopositivity of Olig2, a marker in the nucleus of oligodendrocyte lineage cells (*Figure 2C*). In addition, the MAG signals overlapped with the heminodes at E15 (*Figure 2D*), indicating that myelin sheaths already exist around the paranodal domains at the very early stage of node formation, in agreement with the idea that the early nodes can shift their position dynamically with myelin maturation (*Figure 2E*). These raised the questions as to when the biased nodal spacing pattern emerges during development and whether and how oligodendrocytes contribute to the process.

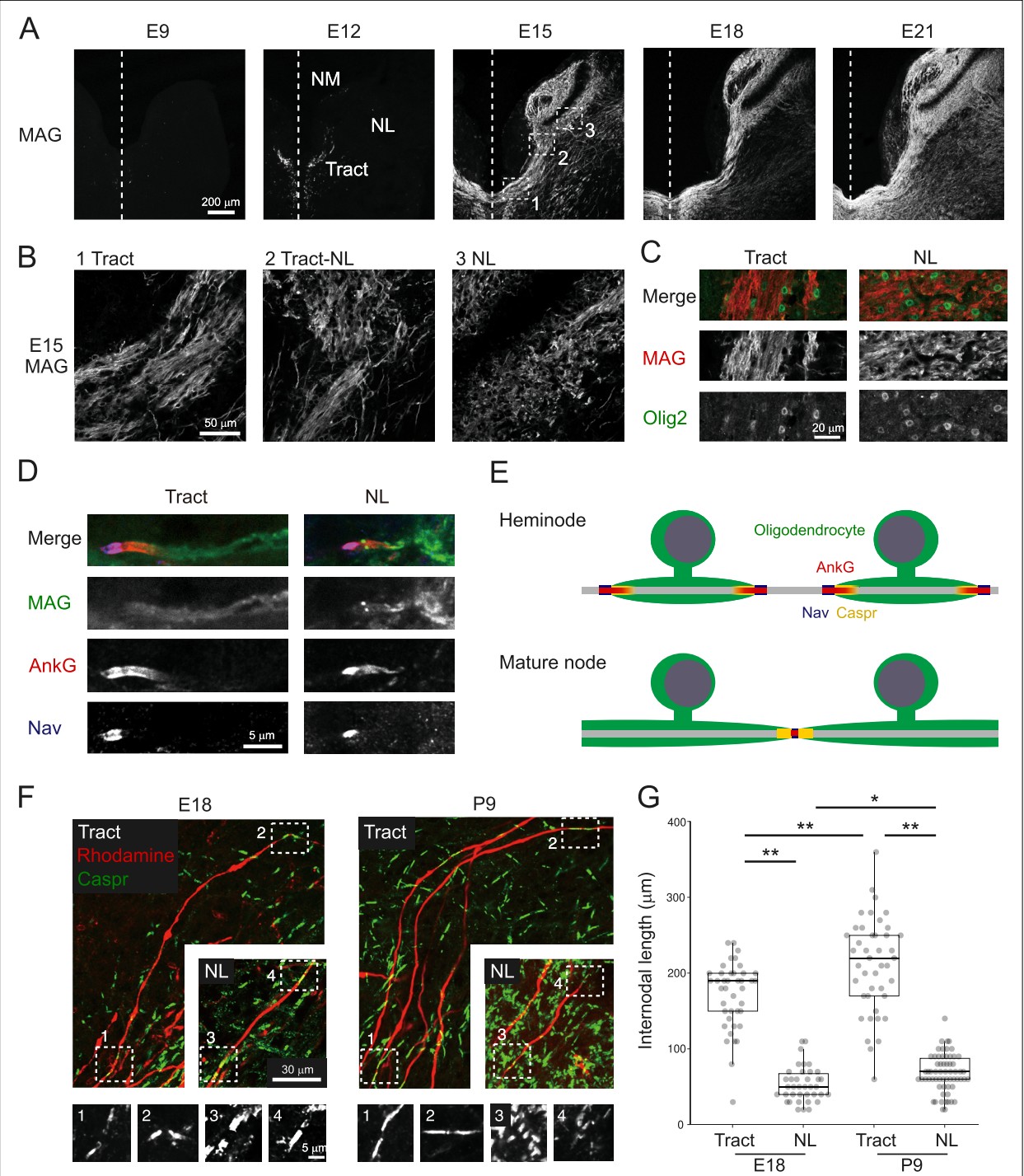

**Figure 2.** Regional differences in internodal length were evident during development. (**A, B**) Immunostaining of MAG between E9 and E21 (**A**). Boxes with numbers at E15 are magnified (**B**). (**C**) Double immunostaining of MAG (red) and Olig2 (green), a marker of oligodendrocyte lineage cells, at E15. (**D**) Localization of MAG (green), AnkG (red), and Nav (blue) on heminode at E15. (**E**) Ranvier node is formed by restricting a gap between adjacent two heminodes during development. Note that paranodal domains, flanking the Nav-positive nodal domain, accompany fibrous AnkG clustering in heminode but not in mature node. (**F, G**) Regional difference in internodal length appeared during the period of node formation. The axons were labeled with rhodamine dextran (red) and stained with Caspr antibody (green) at E18 and P9 (**F**). Internodal length was measured as a distance between adjacent mature/immature nodes (**G**). Boxes with numbers correspond to high-magnification images of each node. E18: n=41 internodes, N=4 chicks for tract, n=38 internodes, N=11 chicks for NL; P9: n=43 internodes, N=7 chicks for tract, n=62 internodes, N=10 chicks for NL. Scale bars: 200 μm (A),

*Figure 2 continued on next page*

*Figure 2 continued*

50 μm (B), 30 μm (F, upper), 20 μm (C), and 5 μm (D and F, lower). Statistical analysis: Kruskal–Wallis test and post hoc Steel–Dwass test (**G**) was used to compare different developmental stages in the same region and different regions at the same developmental stage: *p<0.05, **p<0.01.

The online version of this article includes the following source data and figure supplement(s) for figure 2:

**Source data 1.** Quantitative measurements with associated statistical analyses and effect size visualizations underlying *Figure 2G*.

**Figure supplement 1.** Immunosignals of MAG and AnkG along nucleus magnocellularis (NM) axons appeared concurrently at E13.

## Internodal length was determined roughly during node formation

To answer these questions, we first evaluated the internodal length between mature/immature nodes at E18 and P9 at the tract and NL regions. These two time points correspond to the periods during and after node formation, respectively. We labeled axons with a tracer, rhodamine dextran, visualized nodes with the Caspr antibody, and measured the internodal length at each region (*Figure 2F*).

The internodal length was about three times longer in the tract region compared with the NL region at P9 (tract: 211.1±9.6 μm; NL: 63.9±3.0 μm) (*Figure 2G*), as observed in posthatch animals (*Seidl et al., 2010*). The difference in internodal length in these two regions of the axons was already evident at E18, although the internodal length was shorter by 20% at E18 in both tract and NL regions (tract: 171.0±7.0 μm; NL: 53.4±3.8 μm). The slightly shorter internodal length at E18 would reflect an elongation of the internode during development in accordance with maturation of the brain size. Thus, the nodal spacing pattern was already determined at the early period of node formation, implying that reorganization of internodes would not contribute to the biased internodal length.

## Morphology of oligodendrocytes differed between tract and NL regions

We next explored the contributions of oligodendrocytes to the regional difference in internodal length by evaluating the morphology of individual oligodendrocytes at E21. We sparsely labeled mature oligodendrocytes by expressing tdTomato with a palmitoylation signal (paltdTomato) under MBP promoter using in ovo electroporation and the iOn switch (*Kumamoto et al., 2020*; *Figure 3A and B*), while labeling bilateral NM axons with GFP by transfection of avian adeno-associated virus (A3V) (*Figure 3C*). The 3D morphology of oligodendrocytes was then observed in 200-μm-thick brainstem slices optically cleared with SlowFade Glass mountant (*Figure 3D*; *Videos 1 and 2*). We focused our analyses on oligodendrocytes ensheathing the main trunk of NM axons.

As presumed, length of myelin sheaths differed significantly between the tract and NL regions (tract: 174.1±5.0 μm; NL: 59.7±2.4 μm) (*Figure 3E*), which was consistent with the difference in internodal length between the regions (*Figure 2B*). Importantly, the diameter of the myelin sheath including the axon did not differ between the regions and did not show any correlation with the myelin length (tract: 1.69±0.05 μm; NL: 1.64±0.07 μm) (*Figure 3F and G*). In each oligodendrocyte, the average myelin length was shorter by 3 times (tract: 181.1±7.3 μm NL: 63.6±4.7 μm), the number of myelin sheaths was larger by 2 times (tract: 3.0±0.2; NL: 6.6±0.6), and the total myelin length was shorter by 1.3 times (tract: 527.9±27.2 μm; NL: 394.6±29.7 μm) at the NL region compared to the tract region (*Figure 3H–J*). In addition, the cell body size was 30% smaller in cross-sectional area at the NL region (tract: 78.2±2.7 μm$^2$; NL: 52.5±1.9 μm$^2$) (*Figure 3K*). Thus, oligodendrocyte morphology was heterogeneous, with distinct differences in several aspects between the tract and NL regions (*Figure 3L and M*), even though they were myelinating the same axons.

## Axon structure did not contribute to the regional difference in nodal spacing

The distinct oligodendrocyte morphologies imply differences in their intrinsic properties (e.g., the ability to produce and extend myelin sheaths), but the involvement of other external constraints cannot yet be ruled out. The structure of axons, such as branch points and diameter, is one of these external constraints and can affect myelin extension (*Murray and Blakemore, 1980*; *Hildebrand et al., 1993*; *Ibrahim et al., 1995*; *Bechler et al., 2015*; *Stedehouder et al., 2019*). We therefore investigated the relationship between the structural features of NM axons and the nodal spacing at E21. To accomplish this, we labeled the axon of NM neurons by expressing GFP with a palmitoylation signal (palGFP) using in ovo electroporation and examined the nodal spacing three-dimensionally along the main trunk of

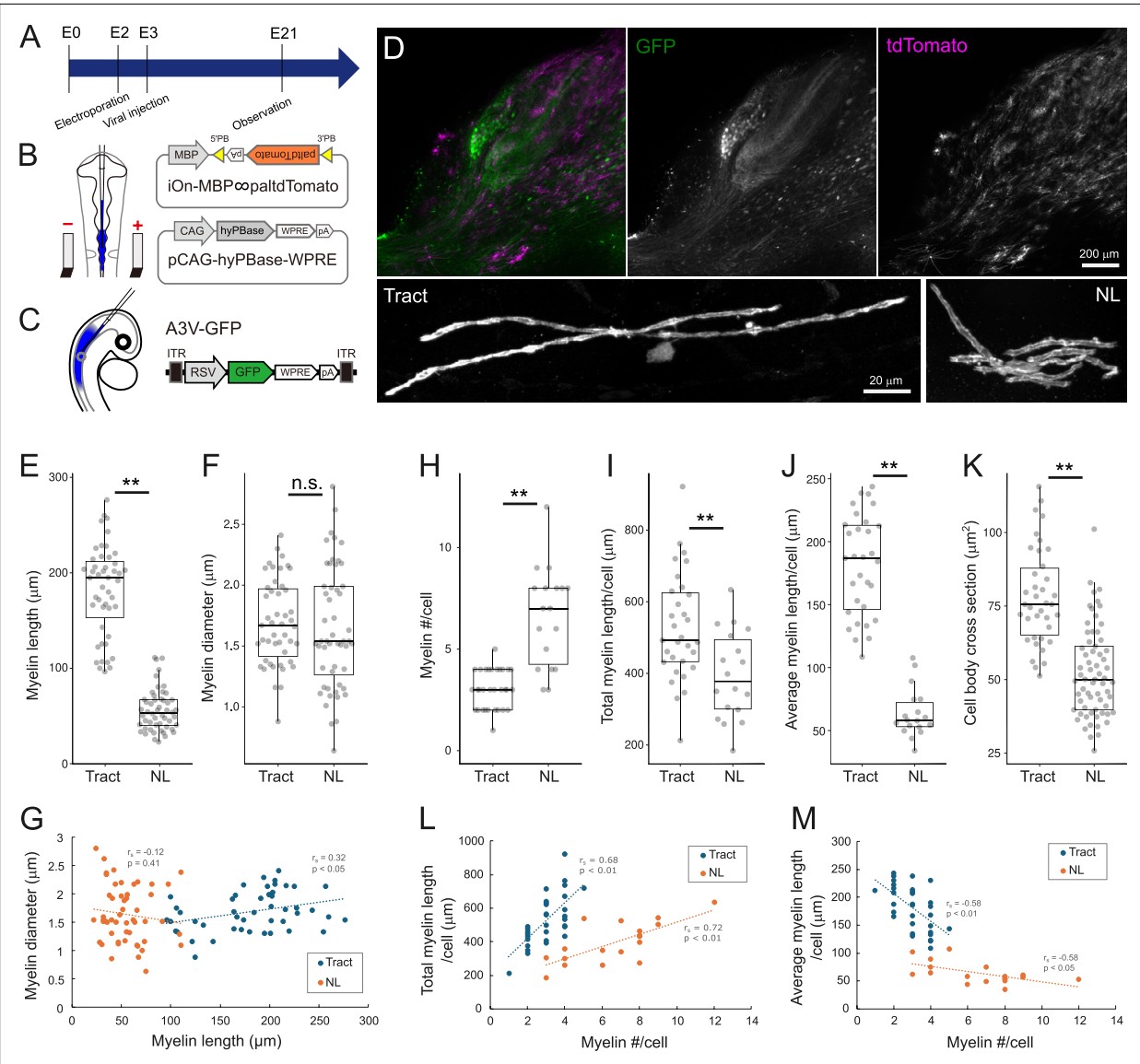

**Figure 3.** 3D morphometry of oligodendrocytes showed distinct differences between tract and nucleus laminaris (NL) regions. (**A**) Timeline of experiments. In ovo electroporation and A3V transfection were performed at E2 (HH stage 11–12) and E2–3 (HH stage 15–16), respectively, and brainstem was observed at E21. (**B**) Two types of plasmid vectors were introduced to visualize mature oligodendrocytes using in ovo electroporation. iOn-MBP∞paltdTomato expresses tdTomato with a palmitoylation signal (paltdTomato) under the control of the MBP promoter after the genomic integration. pCAG-hyPBase integrates the above plasmid into the genome by expressing hyperactive piggyBac transposase. (**C**) A3V-GFP was injected into neural tube to visualize the axon of nucleus magnocellularis (NM) neurons. (**D**) GFP and paltdTomato expressions in a 200-μm-thick slice (upper; scale bars is 200 μm) and magnified images of single oligodendrocytes at tract and NL regions (lower; scale bars is 20 μm). NM neurons and their axons were densely labeled with GFP (green), while mature oligodendrocytes were sparsely labeled with paltdTomato (magenta). (**E–G**) Myelin morphometry showed concordance with internodal length and their uncorrelation with myelin diameter. Length (**E**) and diameter (**F**) of myelin sheaths were compared between regions, and their relationship (**G**) was evaluated with Spearman's rank correlation coefficient ($r_s$). These parameters did not correlate with each other at both regions. Tract: n=47 myelins, N=3 chicks; NL: n=52 myelins, N=3 chicks. (**H–M**) Oligodendrocyte morphometry highlighted their regional heterogeneity. Number (**H**), total length (**I**), average length of myelin sheaths (**J**) per oligodendrocyte, and cross-sectional area of cell body (**K**) were compared between regions. The relationship between total (**L**) or average (**M**) myelin lengths and the number of myelin sheaths was evaluated with Spearman's rank correlation coefficient ($r_s$). Total and average myelin lengths showed increasing and decreasing tendencies, respectively, with an increase of myelin processes. (**H, I, J, L, M**) n=31 cells, N=4 chicks for tract, n=18 cells, N=3 chicks for NL. (**K**) n=38 cells, N=3 chicks for tract, n=65 cells, N=3 chicks for NL. Statistical analysis: Wilcoxon rank sum test (**E, H, J, K**), two-tailed Welch's *t*-test (**F**), and two-tailed Student's *t*-test (**I**): *p<0.05, **p<0.01, n.s., not significant.

The online version of this article includes the following source data for figure 3:

**Source data 1.** Quantitative measurements with associated statistical analyses and effect size visualizations underlying *Figure 3E-M*.

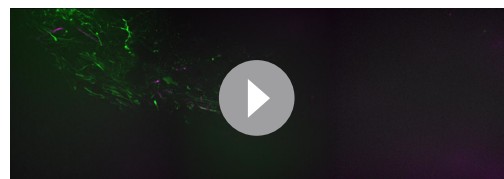

**Video 1.** High-resolution 3D serial images of oligodendrocytes at the tract region of a 200-µm-thick brainstem slice. NM axons were densely labeled with GFP (green) and mature oligodendrocytes (magenta) were sparsely labeled with paltdTomato. The three z-stack images were stitched together. Field of view: 614.89 × 222.08 × 180.9 µm.

https://elifesciences.org/articles/106415/figures#video1

NM axon in 200-µm-thick brainstem slices, which were optically cleared after staining with a Caspr antibody (*Figure 4A–D*; *Video 3*).

The results showed that on the main trunk of the axons, most of the Caspr signals showed mature/immature node patterns, confirming that the axons were almost fully myelinated. These axons formed multiple collateral branches innervating the ventral dendritic layer at the NL region, and triads of paranodal domains were frequently observed at the branch points. The internodal length (73.1±2.1 µm) was about 40% shorter compared to the branch point interval (122.5±6.7 µm) at the NL region (*Figure 4E and F*). Correspondingly, the number of internodes between adjacent branch points averaged 1.6 and was correlated with the length of branch point interval ('0': 27.5±3.9 µm; '1': 79.3±5.1 µm; '2': 156.3±7.1 µm; '3≦': 232.0±10.0 µm) (*Figure 4H and I*). In areas with very short branch point intervals (bottom 10%:<40 µm), the axon lacked Caspr signals frequently (61.5%), implying that these areas are less likely to be myelinated (internode = 0 µm) (*Figure 4G*). On the other hand, even in areas with long branch point intervals (top 10%: >220 µm), the internodal length did not increase and was two times shorter than that at the tract region (<220 µm: 71.4±2.4 µm; >220 µm: 79.9±3.9 µm; tract: 164.3±4.2 µm) (*Figure 4J*). These observations suggest that the branch point interval is not a critical determinant of the short internodes at the NL region, although it can be a limiting factor in areas with the shorter branch point interval.

The diameter of the axon main trunk did not differ between the tract and NL regions (tract: 1.40±0.03 µm; NL: 1.47±0.04 µm) (*Figure 4K*), corresponding to the diameter of myelin sheaths including axon (*Figure 3F*). This was consistent with the previous report showing that the axon diameter did not differ between the tract and the distal portion of ventral NL, at which the axon main trunk is situated (*Seidl et al., 2010*), confirming that the diameter of NM axons is not a factor determining the regional differences in nodal spacing.

## Oligodendrogenesis was augmented at the NL region

Given that NM axons were fully myelinated, oligodendrocyte density could also be a constraint of individual oligodendrocyte morphology; if the density is high, it may limit the extension of myelin sheath due to mutual competition for territory on the axon. Therefore, we examined the density of oligodendrocyte lineage cells during development by labeling the cells with the Olig2 antibody and quantifying their density at the tract and NL regions between E9 and E21 (*Figure 5A* top, B).

The results showed that the density of Olig2-positive cells increased with development, peaked at E15, and decreased slightly afterwards. Importantly, the extent of this increase was more robust and about 1.3-fold higher at the NL region than at the tract region. The differences in oligodendrocyte density could result from several factors, including the density of oligodendrocyte precursor cells (OPCs), which mature into oligodendrocytes through several steps of maturation, as well as the migration, proliferation, and differentiation of these cells. We measured the density of OPCs by labeling the cells with Nkx2.2 antibodies (*Figure 5A* bottom, C; *Xu et al., 2000*). The density of Nkx2.2-positive cells increased sharply and reached a plateau by E12 at both regions, while the extent of this increase was also higher at the NL region. These cells were restricted to the ventral and dorsal fiber layers and almost absent in the somatic and dendritic layers of NL. These

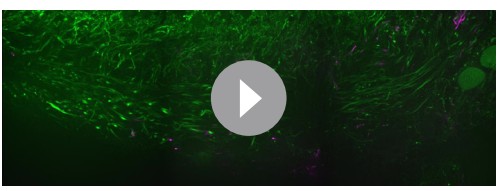

**Video 2.** High-resolution 3D serial images of oligodendrocytes at the NL region of a 200-µm-thick brainstem slice. NM axons were densely labeled with GFP (green) and mature oligodendrocytes (magenta) were sparsely labeled with paltdTomato. The three z-stack images were stitched together. Field of view: 626.58 × 222.08 × 184.5 µm.

https://elifesciences.org/articles/106415/figures#video2

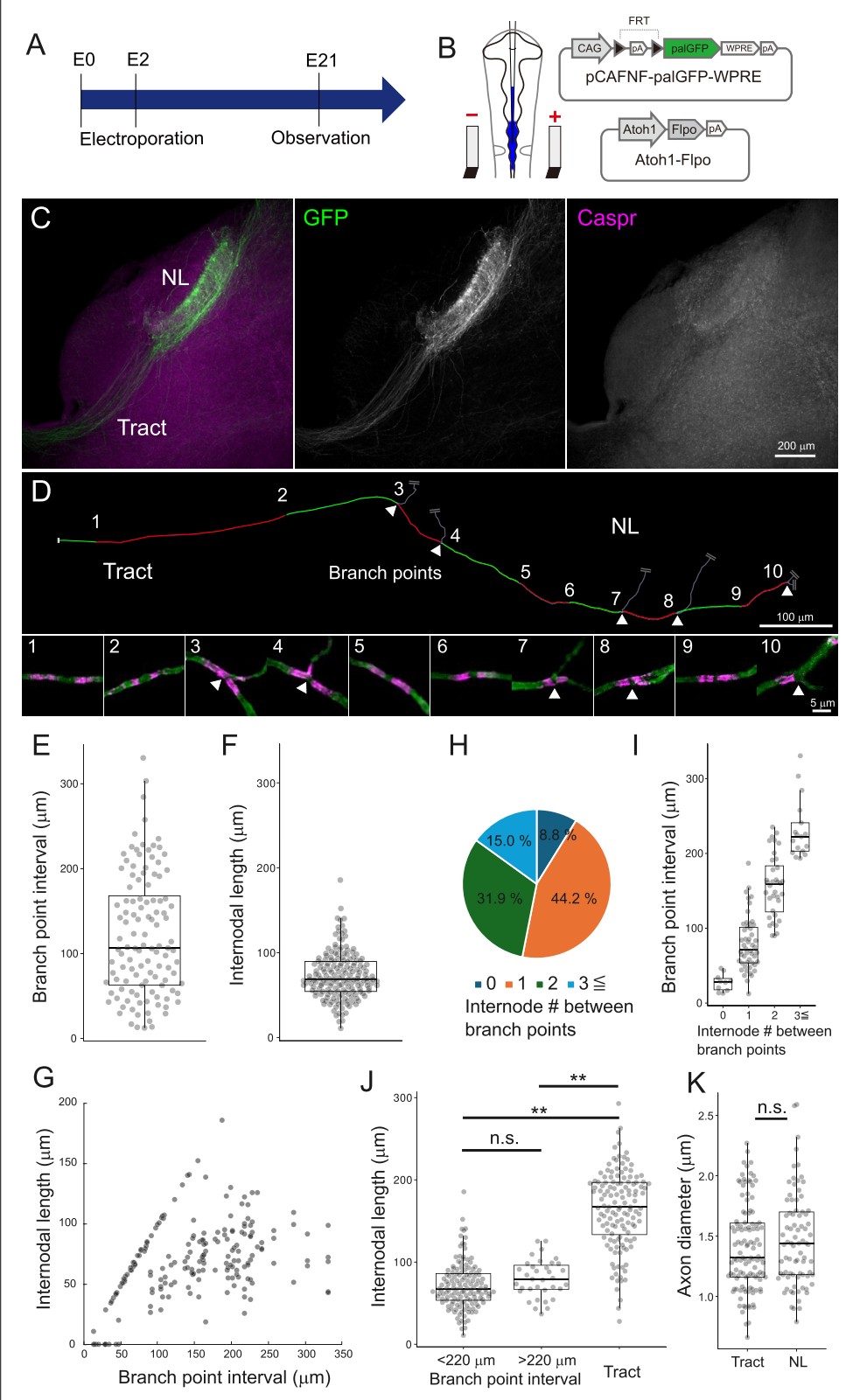

**Figure 4.** Axonal structure was not a major determinant of regional difference in nodal spacing. (**A**) Timeline of experiments. In ovo electroporation was performed at E2 (HH stage 11–12), and brainstem was observed at E21. (**B**) Two types of plasmid vectors were introduced to visualize the axon of nucleus magnocellularis (NM) neurons using in ovo electroporation. Atoh1-Flpo expresses Flpo under the control of Atoh1 promoter, which has selectivity

*Figure 4 continued on next page*

*Figure 4 continued*

for NM neurons. pCAFNF-palGFP-WPRE expresses GFP with a palmitoylation signal (palGFP) in a Flpo-dependent manner. (**C**) Contralateral projections of NM neurons were labeled with palGFP (green), while visualizing paranodes with Caspr antibody (magenta) in a 200-µm-thick slice. (**D**) Nodal spacing across tract and nucleus laminaris (NL) regions along a single NM axon. Arrowheads indicate branch points of collaterals. Each number corresponds to high-magnification images of node (lower panels). Each internode (between nodes) on the axon was labeled alternately with red and green lines, and their lengths were measured. (**E–G**) Internodal length was clearly shorter than the branch point interval at the NL region. Branch point interval (**E**) at the NL region, internodal length (**F**) within those intervals, and their relationship (**G**). Note that the internodal length was mostly below 100 µm at the NL region even when the branch point interval was above 100 µm. (**E**) n=114 intervals, (**F**) n=178 internodes, (**G**) n=188 internodes (including 0 µm), N=6 chicks. (**H–I**) Branch point interval correlated with the number of internodes within the interval. Number of internodes between adjacent branch points (**H**) and branch point interval against the number of internodes (**I**). '0': n=10 intervals; '1': n=50 intervals; '2': n=36 intervals; '3≦': n=17 intervals. (**J**) Internodal length was not necessarily specified by branch point interval. Comparison of internodal length for branch point intervals above 220 µm (top 10% of measured values), below 220 µm at the NL region, and at the tract region. <220: n=143 internodes, >220: n=35 internodes; tract: n=132 internodes, N=5 chicks. (**K**) Diameter of main trunk of the axon at tract and NL regions was not different. Tract: n=114 axons, N=3 chicks; NL: n=81 axons, N=3 chicks. Scale bars: 200 µm (C), 100 µm (D, upper) and 5 µm (D, lower). Statistical analysis: Kruskal–Wallis test and post hoc Steel–Dwass test (**J**) and Wilcoxon rank sum test (**K**): *$p<0.05$, **$p<0.01$, n.s., not significant.

The online version of this article includes the following source data for figure 4:

**Source data 1.** Quantitative measurements with associated statistical analyses and effect size visualizations underlying *Figure 4E-K*.

results suggest that the density of OPCs increases around the period of hearing onset to a larger extent at the NL region, contributing to the higher density of oligodendrocyte lineage cells at this region. We evaluated the level of proliferation of OPCs with BrdU labeling and found that the density of BrdU-positive cells was significantly higher at the NL region during E12–16 (*Figure 5D and E*). BrdU-positive cells were immunopositive for both Olig2 and Nkx2.2, confirming that they were OPCs (*Figure 5F and G*). These results suggest that the higher densities of OPCs and oligodendrocyte lineage cells at the NL region occur through a region-specific facilitation of oligodendrogenesis.

## Inhibition of vesicular release did not affect internodal length but caused unmyelinated segments at the NL region

Myelination could be modulated adaptively by neuronal activity (*Fields, 2015*; *Bechler et al., 2018*; *Bonetto et al., 2021*; *Osanai et al., 2022*). We here tested the possible contributions of activity-dependent adaptive mechanisms to the nodal spacing at E21 by using enhanced tetanus neuro-toxin light chains (eTeNT), a genetically encoded silencing tool inhibiting vesicular release from axons (*Kinoshita et al., 2012*; *Sooksawate et al., 2013*). Using A3V, we expressed GFP-tagged eTeNT in most of the bilateral NM neurons (89.7±1.6%, N=6 chicks) and compared their effects on nodal spacing along these axons between the regions (*Figure 6A–D*).

The results showed that although Caspr-positive nodes were still observed in the axons expressing eTeNT, these axons exhibited an ~30% increase in the fraction of heminodes specifically at the NL region (A3V-GFP: 12.4%; A3V-eTeNT: 41.1%), resulting in the emergence of unmyelinated segments (A3V-GFP: 4.6 ± 1.2%; A3V-eTeNT: 16.0 ± 2.5%) (*Figure 6D–G*). Interestingly, eTeNT expression did not affect the internodal length at both tract and NL regions (*Figure 6H*). The internodal lengths at the NL region did not differ between those with (65.4±2.6 µm) and without (71.7±2.3 µm) heminodes, while the length of unmyelinated segments was only less than half of them (30.7±2.4 µm) (*Figure 6I*). Notably, the increase in the fraction of heminodes and the emergence of unmyelinated segments at the NL region did not occur when the same eTeNT construct was sparsely expressed by in ovo electroporation (heminodes: 14.7%; unmyelinated segments: 2.5 ± 1.2%)

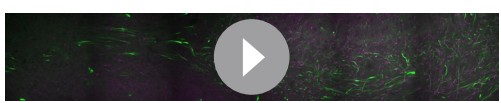

**Video 3.** High-resolution 3D serial images of nodal spacing pattern along NM axon at the tract and NL regions of a 200-µm-thick brainstem slice. NM axons were labeled with GFP (green), and Caspr (magenta) was immunostained. The six z-stack images were stitched together. Field of view: 1237.35 × 217.97 × 185.4 µm.
https://elifesciences.org/articles/106415/figures#video3

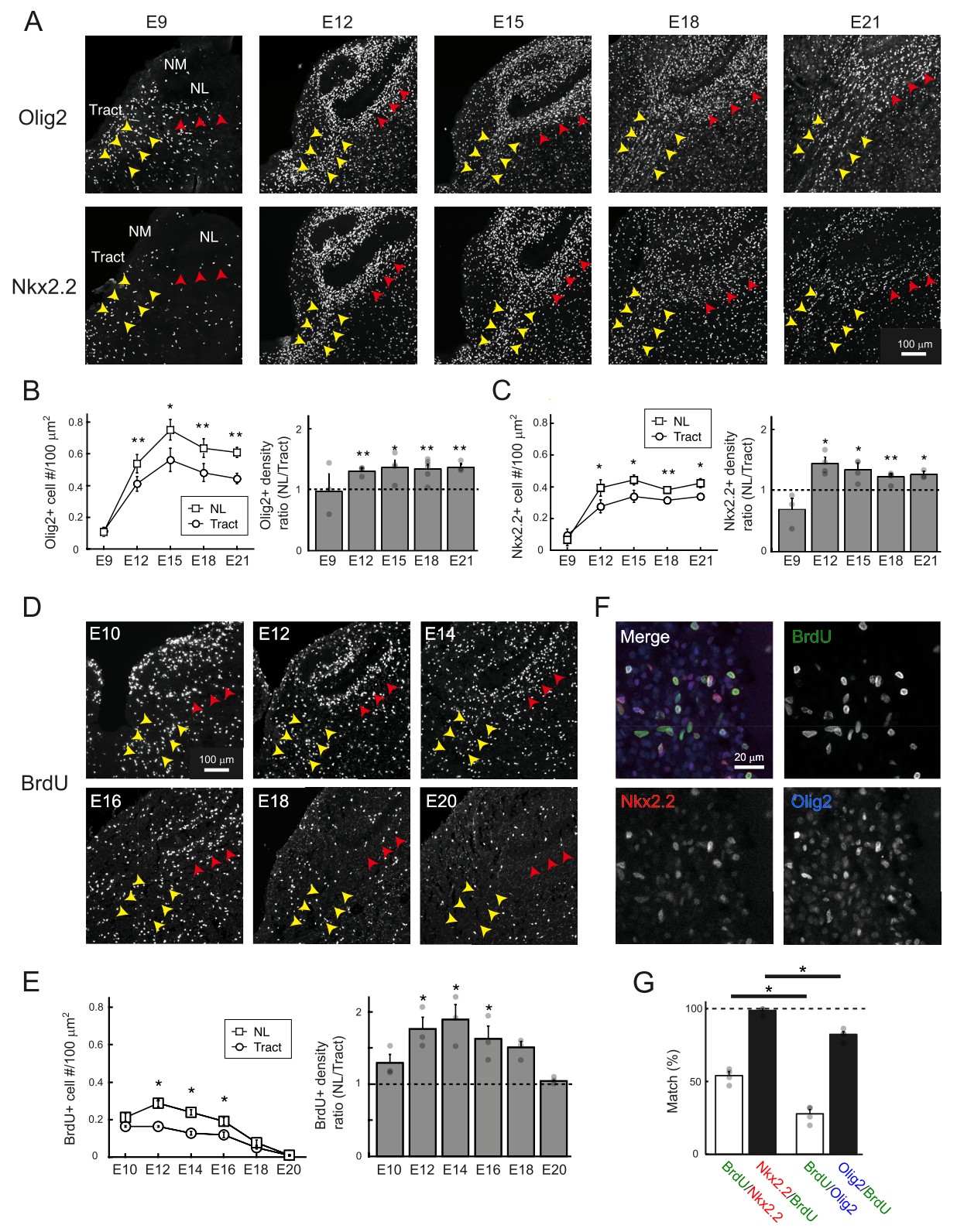

**Figure 5.** Region-specific facilitation of oligodendrogenesis led to higher oligodendrocyte density at the nucleus laminaris (NL) region. (**A**) Immunostainings of Olig2, a marker for oligodendrocyte lineage cells, and Nkx2.2, a marker for oligodendrocyte precursor cells (OPCs). Yellow and red arrowheads indicate tract and NL regions, respectively. (**B, C**) Density of developing oligodendrocytes increased especially at the NL region. Developmental changes in the density of Olig2-positive (**B**) and Nkx2.2-positive (**C**) cells at each region and their ratio between the regions. E9: N=3

*Figure 5 continued on next page*

*Figure 5 continued*

chicks, E12: N=4 chicks, E15: N=4 chicks, E18: N=6 chicks, E21: N=3 chicks. (**D**) Immunostaining of BrdU, a marker for proliferating cells. Yellow and red arrowheads indicate tract and NL regions, respectively. (**E**) Cell proliferation was facilitated at the NL region. Developmental changes in the density of BrdU-positive cells at each region and their ratio between the regions. N=3 chicks for each stage. (**F**) Colocalization of BrdU (green), Nkx2.2 (red), and Olig2 (blue) signals at E12. (**G**) Proliferating cells were almost exclusively OPCs. Mutual percentages between BrdU-positive cells and Nkx2.2- or Olig2-positive cells at E12 and E14. N=4 chicks. Scale bars: 100 μm (**A, D**) and 20 μm (**F**). , (**D**) and 20 μm (**F**). Statistical analysis: Two-tailed paired *t*-test (**B, C, E**), Wilcoxon rank sum test (**G**): *p<0.05, **p<0.01.

The online version of this article includes the following source data for figure 5:

**Source data 1.** Quantitative measurements with associated statistical analyses underlying *Figure 5*.

## Inhibition of vesicular release did not affect oligodendrocyte morphology but suppressed oligodendrogenesis at the NL region

The reduction of myelin sheath density by A3V-eTeNT could occur through a decrease in the number of myelin sheaths formed by each oligodendrocyte and/or through a decrease in the density of oligodendrocytes themselves. To test these possibilities, we sparsely labeled mature oligodendrocytes, while transfecting A3V-eTeNT into bilateral NM neurons, and evaluated the effects of eTeNT on their morphology at E21 (*Figure 6J and K*).

The results showed that eTeNT affected neither the number nor the length of myelin sheaths in each oligodendrocyte at the NL region, supporting the conclusion that the oligodendrocyte morphology is determined in region-specific but activity-independent manners along NM axons.

To determine if inhibiting vesicular release from axons alters proliferation of OPCs, we examined the density of Nkx2.2-positive OPCs and BrdU-positive proliferating cells at E15 in the A3V-eTeNT and control (A3V-GFP) conditions (*Figure 6L and M*). In the control group, the density of these cells became higher by 1.5 folds at the NL region compared to the tract region (see also *Figure 5C and E*). In the eTeNT group, on the other hand, this enhancement was suppressed, which abolished the difference in the density of these cells between the regions. These data indicate that vesicular release from NM axons promotes oligodendrogenesis specifically at the NL region without altering oligodendrocyte morphology, suggesting that adaptive oligodendrogenesis is important in covering the entire length of the axon with short myelin sheaths at the region (*Figure 7A–C*).

## Discussion

This study identified several factors responsible for biased nodal spacing patterns along axons, including heterogeneity in oligodendrocyte morphology, regional differences in proliferation of OPCs, and activity-dependent signaling from axons through vesicular release.

Using the chick brainstem auditory circuit known for its characteristic nodal spacing as a model, we found that NM axons were almost fully myelinated by oligodendrocytes with distinct morphologies between the two regions having markedly different internodal lengths. This regional heterogeneity reflected differences in the intrinsic property of oligodendrocytes in these two regions rather than the extrinsic constraints. Expression of eTeNT in most of the NM axons caused unmyelinated segments by region-specific suppression of oligodendrogenesis without affecting internodal length or oligodendrocyte morphology, suggesting that primary role of neural activity is to ensure the oligodendrocyte density necessary for full myelination of axons.

## Biased nodal spacing pattern reflects regional heterogeneity of oligodendrocyte

The major factor contributing to regional differences in internodal length along NM axons was regional heterogeneity in oligodendrocyte morphology. The morphological differences were observed not only in the length of myelin sheaths, but also in the number of myelin sheaths and the size of cell bodies (*Figure 3*). Considering the morphological classification of oligodendrocytes by *Del Río Hortega,*

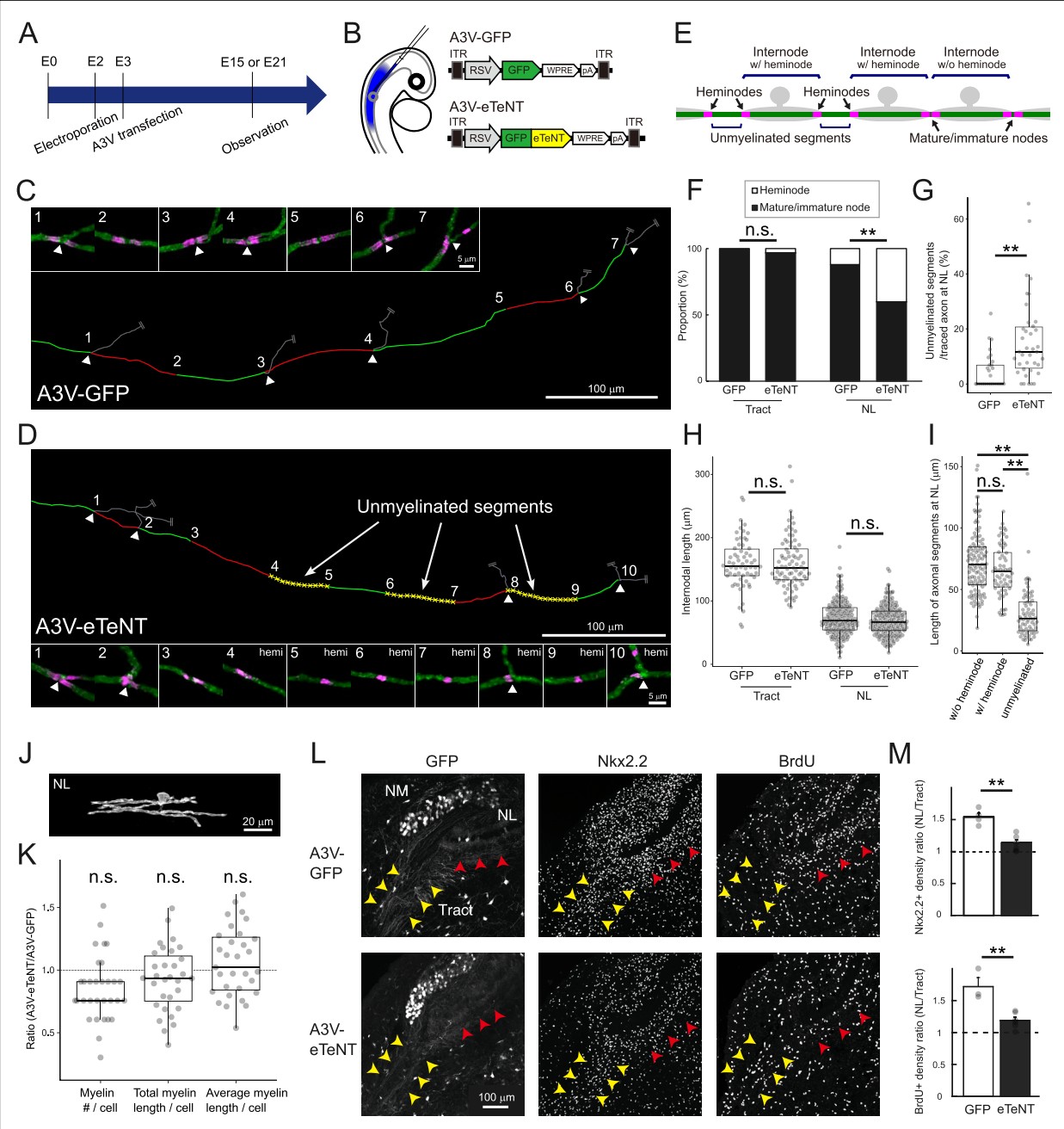

**Figure 6.** Inhibition of vesicular release caused unmyelinated segments via suppression of oligodendrogenesis at the nucleus laminaris (NL) region. (**A**) Timeline of experiments. In ovo electroporation and A3V transfection were performed at E2 (HH stage 11–12) and at E2–3 (HH stage 15–16), respectively, and brainstem was observed at E15 in (**L, M**) and at E21 in (**C–K**). (**B**) A3V-eTeNT expressing GFP-tagged eTeNT was used to inhibit vesicular release from NM axons. A3V-GFP expressing only GFP was used as a control. (**C–I**) eTeNT caused unmyelinated segments at the NL region without affecting internodal length. Most of the axons were labeled with A3V-GFP (**C**) or A3V-eTeNT (**D**), while nodal spacing was analyzed along one of the axons. Arrowheads indicate branch points of collaterals. Each number corresponds to high-magnification images of the node. Each internode was labeled alternately with red and green lines, while unmyelinated segments were labeled with yellow broken lines for A3V-eTeNT. 'Unmyelinated segment' was identified as a non-overlapping axonal segment formed by a pair of heminodes, while internodes were classified into 'internode w/o heminode' and 'internode w/ heminode' according to the types of nodes at the ends (**E**) (see 'Materials and methods'). Percentages of heminode and mature/immature node (**F**), and internodal length (**H**) at tract and NL regions. Percentage of unmyelinated segments in each axon traced over 200 µm (**G**), and length of each segment including internode w/o and w/ heminode, and unmyelinated segment for A3V-eTeNT (**I**) at the NL region. (**F**) GFP (n=124 nodes) and eTeNT (n=160 nodes) at tract, GFP (n=186 nodes) and eTeNT (n=270 nodes) at NL. (**G**) n=29 axons for GFP, n=38 axons for eTeNT. (**H**) GFP (n=62 internodes) and eTeNT (n=80 internodes) at tract, GFP (n=178 internodes) and eTeNT (n=180 internodes) at NL. (**I**) Internode w/o (n=115) and w/ (n=63) heminode and unmyelinated segment (n=63). Tract: N=3 chicks for GFP, N=4 chicks for eTeNT, NL: N=5 chicks for GFP, N=5 chicks for eTeNT.

*Figure 6 continued on next page*

*Figure 6 continued*

(**J–K**) eTeNT did not affect oligodendrocyte morphology at the NL region. Magnified images of single oligodendrocytes after A3V-eTeNT transfection (**J**). Number, total length, average length of myelins per oligodendrocyte were compared between A3V-eTeNT and A3V-GFP (***Figure 4H–J***) and shown as a ratio (**K**; n=31 cells, N=4 chicks). (**L–M**) eTeNT suppressed oligodendrogenesis at the NL region. Immunostainings of Nkx2.2 (OPC marker) and BrdU (proliferating cell marker) at E15 for A3V-GFP (upper) and A3V-eTeNT (lower) (**L**). Yellow and red arrowheads indicate tract and NL regions, respectively. Relative density of Nkx2.2- and BrdU-positive cells between the tract and NL regions (**M**). A3V-eTeNT reduced the density of these cells specifically at the NL region and abolished the difference between the regions. (**M**) N=4 chicks for GFP, N=6 chicks for eTeNT. Scale bars: 200 µm (C, D, upper, and L), 20 µm (J) and 5 µm (D, lower). Statistical analysis: Chi-square test (**F**), Wilcoxon rank sum test (**G, H, K**), Kruskal–Wallis test and post hoc Steel–Dwass test (**I**) and two-tailed Student's *t*-test (**M**): *p<0.05, **p<0.01, n.s., not significant.

The online version of this article includes the following source data and figure supplement(s) for figure 6:

**Source data 1.** Quantitative measurements with associated statistical analyses and effect size visualizations underlying ***Figure 6***.

**Figure supplement 1.** Sparse expression of eTeNT did not cause unmyelinated segments.

*1928*, oligodendrocytes at the tract and NL regions could be classified into types III and II, respectively. Such morphological heterogeneity could be related to the variation of gene profiles among oligodendrocytes (***Butt et al., 1998***; ***Marques et al., 2016***; ***Osanai et al., 2022***; ***Valihrach et al., 2022***). Differences in the origin and/or the pericellular microenvironment of OPCs during development may contribute to the diverse gene profiles of oligodendrocytes (***Crawford et al., 2016***; ***Foerster et al., 2019***; ***Foerster et al., 2024***; ***Boshans et al., 2020***; ***Sherafat et al., 2021***). Consistent with the idea, OPCs in the brainstem have two developmental origins, derived from the ventral and dorsal sides of the hindbrain (***Vallstedt et al., 2005***). In addition, pericellular concentration of some ligands may differ between the tract and NL regions according to the distance from synapses or postsynaptic cells. Whether the gene profiles of oligodendrocytes and OPCs differ between the tract and NL regions and what causes the differences are important issues to be examined in the future.

The regional differences in the morphology of oligodendrocytes would primarily reflect their intrinsic properties specific to each region; the ability of myelin production is higher, while that of myelin extension is lower at the NL region than at the tract region. Supportively, the axonal factors, such as structure and vesicular release, did not affect the regional heterogeneity in oligodendrocyte morphology (***Figures 4 and 6H***). Oligodendrocyte density did not affect the morphology, either; the number and length of myelin sheaths did not increase even when unmyelinated segments appeared after A3V-eTeNT transfection (***Figure 6J–K***). Another potential factor influencing the regional differences in internodal length could be pre-nodal clusters which are formed by cell-autonomous accumulation of nodal components on the axons of certain types of neurons before myelination (***Kaplan et al., 1997***; ***Freeman et al., 2015***; ***Vagionitis et al., 2022***). However, our immunostaining results failed to detect such clusters on the NM axons (***Figure 2—figure supplement 1***). This may be due to limited detection sensitivity or the unavailability of neurofascin antibodies in chickens. Nevertheless, it is unlikely that fixed pre-nodal clusters pre-determine internodal length prior to myelination, given the facts that the length of unmyelinated segments was far shorter than the internodal length at the NL region (***Figure 6I***). Thus, we consider that the morphology of oligodendrocytes, and hence the length of myelin sheaths, is determined according to the intrinsic properties of oligodendrocytes at each region, which underlies the regional differentiation of nodal spacing along NM axons. Variations in the intrinsic properties of oligodendrocytes are also reported in other brain regions. Oligodendrocytes in white matter are morphologically distinct from those in gray matter, and their original features did not change even when OPCs were cross-transplanted (***Viganò et al., 2013***). In addition, OPCs collected from different brain regions show their original morphological features when cultured on artificial fibers (***Bechler et al., 2015***). In summary, our findings provide the perspective that oligodendrocytes with different intrinsic properties work together to fine-tune the neural circuit function involved in ITD detection.

## Adaptive oligodendrogenesis ensures full myelination of axons at the NL region

Neuronal activity can affect the proliferation, maturation, and morphology of oligodendrocytes, but the effects are not uniform across different neuronal subtypes and brain regions (***Gibson et al., 2014***; ***Yang et al., 2020***; ***Bonetto et al., 2021***; ***Osanai et al., 2022***). Indeed, the inhibition of vesicular release had different effects on myelination depending on neuronal subtype; myelination is impaired

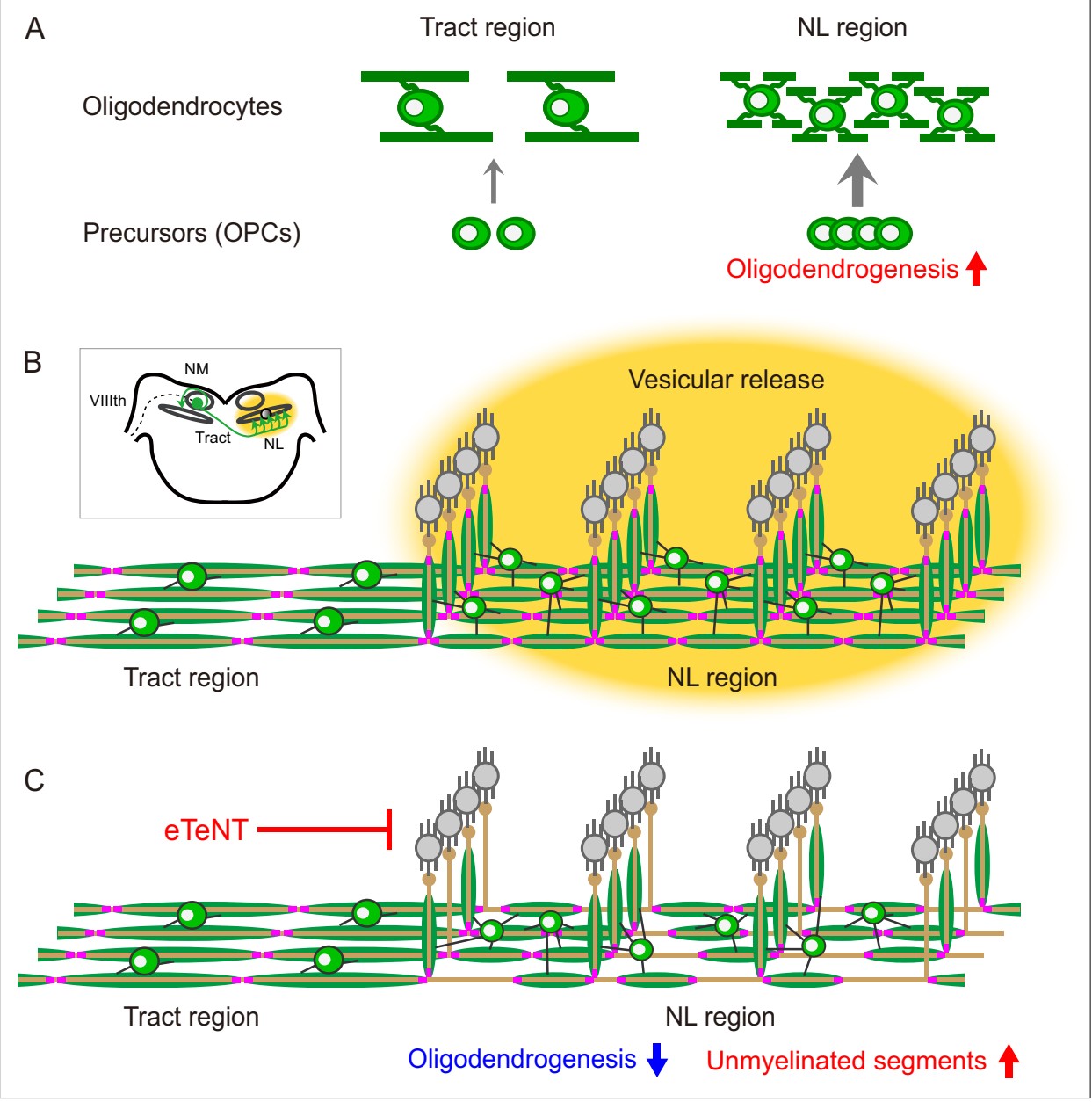

**Figure 7.** Regional heterogeneity of oligodendrocytes and adaptive oligodendrogenesis underlie the biased nodal spacing pattern along nucleus magnocellularis (NM) axons. (**A**) Morphology of oligodendrocytes, such as the number and length of myelins, is determined intrinsically at each region; those at the nucleus laminaris (NL) region have larger numbers of short myelins compared to those at the tract region. In addition, adaptive oligodendrogenesis increases the density of oligodendrocytes specifically at the NL region. (**B, C**) Nodal spacing primarily reflects the length of myelins at each region (**B**). Inhibition of vesicular release from NM axons by eTeNT suppressed adaptive oligodendrogenesis and caused unmyelinated segments at the NL region without altering oligodendrocyte morphology and internodal length (**C**), suggesting the importance of adaptive oligodendrogenesis in myelinating the entire axons with the short myelins at the NL region. Thus, intrinsic and adaptive properties of oligodendrocytes play a pivotal role in shaping the region-specific nodal spacing along NM axons.

in some axons, but remains unaffected in others (*Mensch et al., 2015*; *Koudelka et al., 2016*). In NM axons, activity-dependent signaling did not affect myelin formation in mature oligodendrocytes but instead facilitated proliferation and differentiation of OPCs. Given that this enhancement of oligodendrogenesis occurred specifically at the NL region (*Figures 5 and 6I*) and that unmyelinated segments were increased only when eTeNT was expressed in a large fraction of the axons via A3V (*Figure 6F and G*, *Figure 6—figure supplement 1*), exocytosis from the presynaptic terminals of NM axons would cause the effects by altering the perisynaptic microenvironment. Glutamate and several trophic

factors such as BDNF, which are released from the terminals of NM axons, may act on the OPC directly via spillover from the synapse and/or indirectly via activation of NL neurons or astrocytes (*Du et al., 2003*; *Pease-Raissi and Chan, 2021*; *Fekete and Nishiyama, 2022*; *Pivoňková et al., 2024*). Supportively, mRNA of NR1, a major subunit of NMDA receptor, is strongly expressed in glial cells, presumably oligodendrocyte lineage cells, at the ventral and dorsal dendritic layers of NL after E10 (*Tang and Carr, 2007*). Future studies are needed to determine what ligands mediate the differentiation and proliferation of OPCs as the activity-dependent signaling at the NL region and whether there are regional differences in the sensitivity of these OPCs to the ligands along the NM axon.

Adaptive oligodendrogenesis would support the function of brainstem auditory circuits by ensuring the full myelination of NM axons. Full myelination of the auditory pathway enables rapid membrane responses and high metabolic support during high-frequency firing, which are critical for precise and reliable processing of sound information (*Kim et al., 2013*; *Moore et al., 2020*). Moreover, the full myelination at the NL region, which is mediated by the adaptive enhancement of oligodendrogenesis, would be particularly important for ITD detection. The contralateral projections of NM neurons have a sequential branching structure called a 'delay line' at the ventral area of NL in chickens. This delay line produces an output delay of up to 180 μs for contralateral NL neurons along the medial to lateral axis (*Overholt et al., 1992*). This delay covers the physiological ITD range for the chicken, allowing the ITDs to be encoded in a 'place map' within the NL (*Jeffress, 1948*; *Köppl and Carr, 2008*). This should provide the neural basis for the ability of excellent sound localization in chickens, which can discriminate ITDs of about 20 μs (8.9° minimum audible angle) (*Krumm et al., 2022*). Reliable myelination of the delay line ensures constant conduction velocity and output delay and hence is critical for precise ITD calculation. Thus, the adaptive oligodendrogenesis at the NL region would be a mechanism that secures the accuracy of ITD calculation in the avian auditory circuit. Interestingly, ITD representation within the NL exhibits plasticity around the period of hearing onset in the barn owl (*Carr et al., 2024*), which coincides with the period of adaptive oligodendrogenesis, speculating its contribution to the plasticity.

## Commonalities and differences with the sound localization circuit in mammals

Regional regulation of conduction velocity has also been reported in mammals. In a brainstem auditory circuit of gerbils, which is involved in the processing of ITDs, several regional differentiations have been observed in the nodal spacing along the axon. In the axon of spherical bushy cells, a mammalian homologue of NM neurons, the internodal length is longer for contralateral side than for the ipsilateral side, presumably compensating for the different axonal pathlength between the two sides (*Seidl and Rubel, 2016*). On the other hand, in the axon of globular bushy cells that project to the medial nucleus of the trapezoid body, the internodal length is longer for those innervating more medial location (i.e., more distal location), ensuring simultaneous inputs within the nucleus (*Ford et al., 2015*). These regional differentiations of nodal spacing would be acquired independently in mammals and birds through evolutionary convergence (*Lipovsek and Wingate, 2018*), emphasizing their importance in securing ITD computation in the microsecond order. Notably, in globular bushy cells, neural activity did not alter the nodal spacing, but instead regulated the conduction velocity by adaptively changing the axon diameter (*Nabel et al., 2024*), suggesting that the mechanism of action of activity-dependent signaling is different among neurons and species. It will be interesting to examine whether regional heterogeneity in the intrinsic properties of oligodendrocytes is a prominent determinant in the biased nodal spacing pattern in mammalian ITD circuits as well. Such comparisons will help to understand the fundamental logic and mechanisms of determining nodal spacing patterns along the axons.

## Conclusion

This study identified the factors mediating the regional differentiation of nodal spacing along single axons in the auditory pathway of chicken; the differentiation of nodal spacing reflects regional heterogeneity in oligodendrocyte morphology, and the activity-dependent signaling from axons via vesicular release contributes to the full myelination of axons through region-specific enhancement of oligodendrogenesis. The results in this model system provide evidence that oligodendrocyte heterogeneity,

which is widely observed in the brain, can contribute to achieving optimal timing of signals and optimal function of local circuits in the nervous system.

# Materials and methods

**Key resources table**

| Reagent type (species) or resource | Designation | Source or reference | Identifiers | Additional information |
|---|---|---|---|---|
| Antibody | pan Nav antibody (mouse monoclonal) | Sigma-Aldrich | Cat# S8809; RRID:AB_477552 | 5 µg/ml |
| Antibody | pan Nav antibody (guinea pig polyclonal) | Hiroshi Kuba, **Kuba et al., 2006** | | 5 µg/ml |
| Antibody | AnkyrinG antibody (rabbit polyclonal) | Gift from Gisèle Alcaraz, **Bouzidi et al., 2002** | | 5 µg/ml |
| Antibody | Caspr antibody (mouse monoclonal) | NeuroMab | Cat# K65/35; RRID:AB_2877274 | 2 µg/ml |
| Antibody | MAG antibody (mouse monoclonal) | Merck | Cat# MAB1567; RRID:AB_11214010 | 2 µg/ml |
| Antibody | Olig2 antibody (rabbit polyclonal) | Gift from Hirohide Takebayashi, **Takebayashi et al., 2000** | | 2 µg/ml |
| Antibody | Nkx2.2 antibody (mouse monoclonal) | DSHB | Cat# 74.5A5; AB_2738924 | 2 µg/ml |
| Antibody | BrdU antibody (rat monoclonal) | Abcam | Cat# ab6326; RRID:AB_305426 | 10 µg/ml |
| Antibody | GFP antibody (rabbit polyclonal) | MBL | Cat# 598; RRID:AB_591816 | 2 µg/ml |
| Antibody | GFP antibody (rat monoclonal) | Santa Cruz Biotechnology | Cat# sc-101536; RRID:AB_1124404 | 2 µg/ml |
| Antibody | RFP antibody (rabbit polyclonal) | Rockland Immunochemicals | Cat# 600-401-379; RRID:AB_2209751 | 2 µg/ml |
| Antibody | Alexa-conjugated secondary antibodies (goat polyclonal) | Thermo Fisher Scientific | Various | 10 µg/ml |
| Chemical compound, drug | Isoflurane | FUJIFILM Wako | Cat# 099-06571 | |
| Chemical compound, drug | Dextran (MW 3000) conjugated with TMR | Life Technologies | Cat# D3307 | 10–40% in 0.1 M phosphate buffer adjusted to pH 2.0 with HCl |
| Chemical compound, drug | BrdU | Nacalai Tesque | Cat# 08779-61 | 10 mg/ml in PBS |
| Chemical compound, drug | SlowFade Glass Soft-set Antifade Mountant | Thermo Fisher Scientific | Cat# S36917 | |
| Commercial assay or kit | In-Fusion Snap Assembly Master Mix | Takara | Cat# 638947 | |
| Commercial assay or kit | NEBuilder HiFi DNA Assembly Master Mix | NEB | Cat# E2621S | |
| Recombinant DNA reagent | iOn-MBP∞paltdTomato | This paper | | See 'Plasmids' section for construction |
| Recombinant DNA reagent | pCAG-hyPBase | This paper | | See 'Plasmids' section for construction |
| Recombinant DNA reagent | pCAFNF-palGFP-WPRE | This paper | | See 'Plasmids' section for construction |
| Recombinant DNA reagent | Atoh1-Flpo | **Lipovsek and Wingate, 2018** | | |
| Recombinant DNA reagent | pCMV-hyPBase | **Yusa et al., 2011** | | |

*Continued on next page*

*Continued*

| Reagent type (species) or resource | Designation | Source or reference | Identifiers | Additional information |
|---|---|---|---|---|
| Recombinant DNA reagent | pCAG-EGFP-WPRE | *Egawa and Yawo, 2019* | | |
| Recombinant DNA reagent | pCAG-floxedSTOP-tdTomato-WPRE | *Egawa and Yawo, 2019* | | |
| Recombinant DNA reagent | pCAFNF-GFP | *Matsuda and Cepko, 2007* | Addgene #13772; RRID:Addgene_13772 | |
| Recombinant DNA reagent | iOn-CAG∞MCS | *Kumamoto et al., 2020* | Addgene #154013; RRID:Addgene_154013 | |
| Recombinant DNA reagent | pAAV2 SynTetOff-palGFP | *Sohn et al., 2017* | | |
| Recombinant DNA reagent | pIP200 containing 1.9 kb sequence of mouse MBP promoter | Gft from Yasuyuki Osanai | | |
| Recombinant DNA reagent | pA3V-RSV-EGFP | *Matsui et al., 2012* | | |
| Recombinant DNA reagent | pA3V-RSV-EGFP.eTeNT | This paper | | see 'Plasmids' section for construction |
| Recombinant DNA reagent | HiRet-TRE-EGFP.eTeNT | *Kinoshita et al., 2012* | | |
| Recombinant DNA reagent | A3V-EGFP | *Matsui et al., 2012* | | Avian adeno-associated virus (A3V) vector prepared from pA3V-RSV-EGFP ($1\times10^{13}$ GC/ml) |
| Recombinant DNA reagent | A3V-eTeNT | This paper | | Avian adeno-associated virus (A3V) vector prepared from pA3V-RSV-EGFP.eTeNT ($1\times10^{13}$ GC/ml) |
| Software, algorithm | Imaris Stitcher | Oxford Instruments | https://imaris.oxinst.com/ | |
| Software, algorithm | SNT | *Arshadi et al., 2021* | https://imagej.net/plugins/snt/ | |
| Software, algorithm | Fiji (ImageJ) | *Schindelin et al., 2012* | https://imagej.net/software/fiji/ | |
| Software, algorithm | Inkscape | The Inkscape Project | https://inkscape.org/ | |
| Software, algorithm | PlotsOfData | *Postma and Goedhart, 2019* | https://huygens.science.uva.nl/ | |
| Software, algorithm | PlotsOfDifferences | *Goedhart, 2019* | https://huygens.science.uva.nl/ | |
| Software, algorithm | Excel | Microsoft | https://www.microsoft.com/ | |
| Software, algorithm | R | R Project | http://cran.r-project.org/ | |

## Animals

Chickens (*Gallus domesticus*) of either sex between embryonic day 9 (E9) and posthatch day 9 (P9) were used for experiments. The care of experimental animals was in accordance with the regulations on animal experiments at Nagoya University and the experiments were approved by the institutional committee. Fertilized eggs were incubated in a humidified incubator at 37.5°C to desired stage. The developmental stage of embryos was determined according to the *Hamburger and Hamilton, 1951* series. Chicks were deeply anesthetized with isoflurane (FUJIFILM Wako), and embryos were anesthetized by cooling eggs in ice-cold water. Brainstem tissues containing the middle one-third of NM were mostly used for experiments (*Kuba et al., 2005*).

## Immunohistochemistry

Mouse pan Nav antibody (5 µg/ml, Sigma-Aldrich), guinea pig pan Nav antibody (5 µg/ml; *Kuba et al., 2006*), rabbit AnkG antibody (5 µg/ml, a gift from Gisèle Alcaraz, *Bouzidi et al., 2002*), mouse Caspr antibody (2 µg/ml, NeuroMab), mouse MAG antibody (2 µg/ml, Merck), rabbit Olig2 antibody (2 µg/ml, a gift from Hirohide Takebayashi, *Takebayashi et al., 2000*), mouse Nkx2.2 antibody (2 µg/ml, DSHB), rat BrdU antibody (10 µg/ml, Abcam), rabbit GFP antibody (2 µg/ml, MBL), rat GFP antibody (2 µg/ml, Santa Cruz Biotechnology), and rabbit RFP antibody (2 µg/ml, Rockland Immunochemicals) were used for immunohistochemistry. Chicks were perfused transcardially with a periodate-lysine-paraformaldehyde fixative (ml/g body weight): 1% (w/v) paraformaldehyde, 2.7% (w/v) lysine HCl, 0.21% (w/v) NaIO$_4$, and 0.1% (w/v) Na$_2$HPO$_4$. The brainstem was postfixed for 1.5 hours at 4°C. After cryoprotection with 30% (w/w) sucrose in PBS, coronal sections (20–30 µm) were obtained. The sections were incubated overnight with the primary antibodies, then with Alexa-conjugated secondary antibodies (10 µg/ml, Thermo Fisher Scientific) for 2 hours and were observed under a confocal laser-scanning microscope (FV1000, Olympus). For each image, 6–9 confocal planes were Z-stacked with a step of 0.8 µm. Images were analyzed using Fiji (*Schindelin et al., 2012*) and assembled using Inkscape (https://inkscape.org/).

Anterograde labeling of NM axons was made by injecting dextran (MW 3000) conjugated with TMR (Life Technologies, 10–40% in 0.1 M phosphate buffer adjusted to pH 2.0 with HCl) into the midline tract region through a patch pipette and incubating the brainstem in the HG-ACSF for 30 minutes at 38°C (*Lawrence and Trussell, 2000*; *Wirth et al., 2008*).

Internodal length was measured as follows. Images were captured with a ×60, 1.35-NA objective (Olympus), and 6–10 confocal planes were Z-stacked with maximum projection. Internodal length was defined as a distance between adjacent nodes, which was determined by identifying a pair of Caspr signals. The number of oligodendrocytes and their precursors was measured as follows. Images were captured with a ×20, 0.75-NA objective (Olympus), and a single confocal plane was used for the measurement. The number was measured automatically by setting a threshold in areas over 14,000 µm$^2$ at each region.

## BrdU labeling

BrdU (10 mg/ml in PBS, Nacalai Tesque) was injected subcutaneously into embryos (0.1 mg/g body weight) 1 hour before fixation, and sections were prepared as described above. The sections were mounted on a cover slide, treated with 2 M HCl for 40 min at 60°C, and used for immunohistochemistry.

## Plasmids

The following plasmids were used for in ovo electroporation: iOn-MBP∞paltdTomato, pCAG-hyPBase, pCAFNF-palGFP-WPRE, and Atoh1-Flpo (a gift from Marcela Lipovsek, *Lipovsek and Wingate, 2018*). All plasmids were constructed by inserting the following sequences into the plasmid backbones of iOn-CAG∞MCS (Addgene #154013, *Kumamoto et al., 2020*), pCAG-EGFP-WPRE and pCAG-floxedSTOP-tdTomato-WPRE (*Egawa and Yawo, 2019*) using In-Fusion Snap Assembly Master Mix (Takara) or NEBuilder HiFi DNA Assembly Master Mix (NEB); 1.9 kb sequence of mouse MBP promoter (a gift from Yasuyuki Osanai), hyPBase (pCMV-hyPBase, a gift from Wellcome Trust Sanger Institute; *Yusa et al., 2011*), CAFNF sequence (pCAFNF-GFP, addgene #13772; *Matsuda and Cepko, 2007*), palGFP (pAAV2 SynTetOff-palGFP, a gift from Hiroyuki Hioki; *Sohn et al., 2017*). For A3V production, pA3V-RSV-EGFP and pA3V-RSV-EGFP.eTeNT were used. The latter was also used for in ovo electroporation in *Figure 6—figure supplement 1*. pA3V-RSV-EGFP.eTeNT was constructed by inserting eTeNT sequence (HiRet-TRE-EGFP.eTeNT; *Kinoshita et al., 2012*) into the plasmid back-bones of pA3V-RSV-EGFP (*Matsui et al., 2012*). All constructs were verified by Sanger sequencing.

## In ovo electroporation

Plasmids were introduced into chick embryos in the same manner as previously reported (*Egawa and Yawo, 2019*; *Jahan et al., 2023*). Briefly, plasmid cocktail (0.4–0.5 µg/µl of each) was injected into the neural tube of chick embryos at E2 (HH stage 10–12; *Hamburger and Hamilton, 1951*) and introduced into the right side of the hindbrain (rhombomere 3–8). The electrical pulses were applied using a pair of electrodes (CUY613P1, NEPAGENE) placed in parallel at 2 mm apart. The settings of

the electroporator (NEPA21, NEPAGENE) were as follows: poring pulse: 15 V, 30 ms width, 50 ms interval, 3 pulses, 10% decay and transfer pulse: 5 V, 50 ms width, 50 ms interval, 5 pulses, 40% decay.

## A3V transfection

A3V-GFP and A3V-eTeNT were prepared from pA3V-RSV-EGFP and pA3V-RSV-EGFP.eTeNT as previously reported (*Matsui et al., 2012*). Their yield was approximately $10^{13}$ genome copies (GC)/ml. Viral solution containing 0.05% Fast Green (Nakarai Tesque) was injected into the lumen of neural tube near the ear vesicle at E2–3 (60–70 hours of incubation, HH stage 15–16), at which A3V infects NM neurons with high efficiency (*Matsui et al., 2012*).

## Immunostaining and 3D imaging of 200-µm-thick brainstem slices

Chick embryos were perfused transcardially at E21 with PBS containing 10 U/ml heparin followed by 4% (w/v) paraformaldehyde in PBS. The brainstem was postfixed overnight at 4°C. After cryoprotection with 30% (w/w) sucrose in PBS, coronal sections (200 µm thickness) were obtained. The brainstem slices were delipidated and permeabilized with cold acetone for 3 minutes, washed in PBS containing 0.1% (w/v) $NaBH_4$ for 5 minutes to remove autofluorescence, and stained with primary antibody for 3 days, followed by secondary antibody for 1 day. The stained slices were embedded with SlowFade Glass Soft-set Antifade Mountant (Thermo Fisher Scientific) using 200-µm-thick silicone rubber spacer and No.0 coverslip (CG00C2, ThorLab), thereby making them transparent.

3D images were captured with a ×60, 1.42-NA oil-immersion objective (UPLXAPO60XO, Olympus) or ×100, 1.5-NA oil-immersion objective (UPLAPO100XOHR, Olympus) under a spinning disk confocal laser-scanning microscope (SpinSR10, Olympus). For each image, approximately 600 confocal planes were Z-stacked with a step of 0.3 µm. Multiple 3D images were stitched together using Imaris Stitcher (Oxford Instruments).

## 3D morphometry

In the 3D images captured by SpinSR10, myelin length, internode length, and branch point interval were traced using SNT (*Arshadi et al., 2021*), a Fiji plugin. Oligodendrocytes presumed not myelinating NM axons (e.g., oligodendrocytes that are not present within NM axon bundles or whose myelin sheaths barely along the axon bundle) were excluded from the analysis. For myelin and axon diameters, the length between intensity peaks of membrane-localized fluorescent proteins on a line orthogonal to the axon was measured. For nodal spacing along a single axon, three different axonal segments were defined according to the types of nodes at the ends; 'internode w/o heminode' was a segment between mature/immature nodes, 'internode w/ heminode' was a segment either between a mature/immature node and a heminode, or between heminodes flanking unmyelinated segments when the axon was separated by multiple heminodes, and 'unmyelinated segment' was a non-overlapping segment between heminodes consisting of adjacent internodes. Quantified data were graphed using PlotsOfData (*Postma and Goedhart, 2019*), PlotsOfDifferences (*Goedhart, 2019*) or Excel (Microsoft).

## Statistics

Statistical analysis was performed using R software (http://cran.r-project.org/). Data normality was assessed using the Shapiro–Wilk test. Non-normally distributed data were analyzed using non-parametric tests: the Wilcoxon rank sum test for comparisons between two groups (*Figures 3 E, H, J, K*, *4K*, *5G*, and *6G, H, K*), and the Kruskal–Wallis test with post hoc Steel–Dwass test for comparisons among more than two groups (*Figures 2G, 4J, and 6I*). Normally distributed data were evaluated for equality of variance by Levene's test; two-tailed Student's *t*-test (*Figures 3I and 6M*) was used for equal variance, and two-tailed Welch's *t*-test (*Figure 3F*) for unequal variance. Two-tailed paired *t*-test was used for comparisons of paired data between two groups measured from the same brainstem slice (*Figure 5B, C, and E*). Chi-square test was used to compare the proportions between groups (*Figures 1H and 6F*). Correlations were assessed using Spearman's rank correlation coefficient (*Figure 3G, L, and M*). The significance level was set at 0.05. Data are presented as mean ± standard error of the mean (SEM).

## Acknowledgements

Special thanks to R Douglas Fields and Sian Lewis for their insightful feedback and suggestions on this manuscript. We acknowledge the Division for Medical Research Engineering, Nagoya University Graduate School of Medicine, for usage of SpinSR10, Imaris Stitcher, and NanoDrop 2000. We thank Gisèle Alcaraz and Hirohide Takebayashi for kindly distributing AnkyrinG and Olig2 antibodies. We also thank Kazuhiro Nakamura and Chika Nishimura for cooperation in A3V production. Further thanks are extended to Connie Cepko, Jean Livet, Marcela Lipovsek, Hiroyuki Hioki, Yasuyuki Osanai and Wellcome Trust Sanger Institute for a gift of pCAFNF-GFP (Addgene #13772; *Matsuda and Cepko, 2007*), iOn-CAG∞MCS (Addgene #154013; *Kumamoto et al., 2020*), Atoh1-Flpo (*Lipovsek and Wingate, 2018*), pAAV2 SynTetOff-palGFP, 1.9 kb MBP promoter, and pCMV-hyPBase (*Yusa et al., 2011*) respectively. This work was supported by Grants-in-aid from MEXT KAKENHI (19H04747, 21H02577, 22K19358 and 24H00584 to HK; 17K07039, 20K15915 and 23K05986 to RE) and the Takeda Science Foundation to HK.

## Additional information

### Funding

| Funder | Grant reference number | Author |
| --- | --- | --- |
| Japan Science and Technology Agency | 17K07039 | Ryo Egawa |
| Japan Science and Technology Agency | 20K15915 | Ryo Egawa |
| Japan Science and Technology Agency | 23K05986 | Ryo Egawa |
| Japan Science and Technology Agency | 19H04747 | Hiroshi Kuba |
| Japan Science and Technology Agency | 21H02577 | Hiroshi Kuba |
| Japan Science and Technology Agency | 22K19358 | Hiroshi Kuba |
| Japan Science and Technology Agency | 24H00584 | Hiroshi Kuba |

The funders had no role in study design, data collection and interpretation, or the decision to submit the work for publication.

### Author contributions

Ryo Egawa, Conceptualization, Data curation, Formal analysis, Funding acquisition, Validation, Investigation, Visualization, Methodology, Writing - original draft, Project administration, Writing – review and editing; Kota Hiraga, Investigation; Ryosuke Matsui, Dai Watanabe, Methodology; Hiroshi Kuba, Conceptualization, Formal analysis, Supervision, Funding acquisition, Project administration, Writing – review and editing

### Author ORCIDs

Ryo Egawa  https://orcid.org/0009-0009-8640-1529
Hiroshi Kuba  https://orcid.org/0000-0002-9256-2726

### Ethics

Animal experiments were approved by the Animal Experiment Committee of Nagoya University (Approval number: M240042-001) and performed in accordance with the regulations on animal experiments at Nagoya University. These experimental protocols were carried out in accordance with the Fundamental Guidelines for Proper Conduct of Animal Experiment and Related Activities in Academic Research Institutions (Notice No. 71 of the Ministry of Education, Culture, Sports, Science and Technology, 2006), the Standards relating to the Care and Keeping and Reducing Pain

of Laboratory Animals (Notice of the Ministry of the Environment No. 88 of 2006) and the Standards relating to the Methods of Destruction of Animals (Notice No. 40 of the Prime Minister's Office, 1995).

Reviewer #2 (Public review): https://doi.org/10.7554/eLife.106415.4.sa1
Reviewer #3 (Public review): https://doi.org/10.7554/eLife.106415.4.sa2
Author response https://doi.org/10.7554/eLife.106415.4.sa3

---

# Additional files

## Supplementary files
MDAR checklist

## Data availability
All data generated or analyzed in this study are included in the manuscript and supporting files. Source data files have been provided for figures. Further information and requests for reagents should be directed to and will be fulfilled by the corresponding author, Hiroshi Kuba (kuba@med.nagoya-u. ac.jp).

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
