## [Editor Report · eLife Assessment]

This **important** study uses the delay line axon model in the chick brainstem auditory circuit to examine the interactions between oligodendrocytes and axons in the formation of internodal distances. This is a significant and actively studied topic, and the authors have used this preparation to support the hypothesis that regional heterogeneity in oligodendrocytes underlies the observed variation in internodal length. In a **solid** series of experiments, the authors have used enhanced tetanus neurotoxin light chains, a genetically encoded silencing tool, to inhibit vesicular release from axons and support the hypothesis that regional heterogeneity among oligodendrocytes may underlie the biased nodal spacing pattern in the sound localization circuit.

[Editors' note: this paper was reviewed by Review Commons.]

---

## [Referee Report · Reviewer #2 (Public review)]

Summary:

Egawa et al describe the developmental timeline of the assembly of nodes of Ranvier in the chick brainstem auditory circuit. In this unique system, the spacing between nodes varies significantly in different regions of the same axon from early stages, which the authors suggest is critical for accurate sound localization. Egawa et al set out to determine which factors regulate this differential node spacing. They do this by using immunohistological analyses to test the correlation of node spacing with morphological properties of the axons, and properties of oligodendrocytes, glial cells that wrap axons with the myelin sheaths that flank the nodes of Ranvier. They find that axonal structure does not vary significantly, but that oligodendrocyte density and morphology varies in the different regions traversed by these axons, which suggests this is a key determinant of the region-specific differences in node density and myelin sheath length. They also find that differential oligodendrocyte density is partly determined by secreted neuronal signals, as (presumed) blockage of vesicle fusion with tetanus toxin reduced oligodendrocyte density in the region where it is normally higher. Based on these findings, the authors propose that oligodendrocyte morphology, myelin sheath length, and consequently nodal distribution are primarily determined by intrinsic oligodendrocyte properties rather than neuronal factors such as activity.

Significance:

In our view the study tackles a fundamental question likely to be of interest to a specialized audience of cellular neuroscientists. This descriptive study is suggestive that in the studied system, oligodendrocyte density determines the spacing between nodes of Ranvier, but further manipulations of oligodendrocyte density per se are needed to test this convincingly.

---

## [Referee Report · Reviewer #3 (Public review)]

Summary:

The authors have investigated the myelination pattern along the axons of chick avian cochlear nucleus. It has already been shown that there are regional differences in the internodal length of axons in the nucleus magnocellularis. In the tract region across the midline, internodes are longer than in the nucleus laminaris region. Here the authors suggest that the difference in internodal length is attributed to heterogeneity of oligodendrocytes. In the tract region oligodendrocytes would contribute longer myelin internodes, while oligodendrocytes in the nucleus laminaris region would synthesize shorter myelin internodes. Not only length of myelin internodes differs, but also along the same axon unmyelinated areas between two internodes may vary. This is an interesting contribution since all these differences contribute to differential conduction velocity regulating ipsilateral and contralateral innervation of coincidence detector neurons. However, the demonstration falls rather short of being convincing.

Significance:

The authors suggest that the difference in internodal length is attributed to heterogeneity of oligodendrocytes. In the tract region oligodendrocytes would contribute longer myelin internodes, while oligodendrocytes in the nucleus laminaris region would synthesize shorter myelin internodes. Not only length of myelin internodes differs, but also along the same axon unmyelinated areas between two internodes may vary. This is an interesting contribution since all these differences contribute to differential conduction velocity regulating ipsilateral and contralateral innervation of coincidence detector neurons.

Editors' note: The authors have written an effective rebuttal to the previous round of reviews.

---

## [Author Response]

The following is the authors’ response to the previous reviews

**Reviewer #3:**
Comments on revised version:This revised version is in large improved and the responses to reviewers' comments are generally relevant. However, the response regarding pre-nodes is not satisfactory. I understand that the authors prefer to avoid further experimentations, but I think this is an important point that needs to be clarified. Exploring stages between E12 and E15 are therefore of importance. When carefully examining some of the figures (Fig. 1E or 2D) I think that at E15 they may well be pre-nodes formation prior to myelin deposition, on structure the authors considered to be heminodes. To be convincing they should use double or triple labeling with, in addition to the nodal proteins (ankG and/or Nav pan), a good myelin marker such as antiPLP. The rat monoclonal developed by late Pr Ikenaka would give a sharper staining than the anti MAG they used. (I assume the clone must still be available in Okazaki).

We appreciate your insightful comment regarding the possible presence of pre-nodal clusters along NM axons and your kind suggestion to use the PLP antibody (clone AA3; Yamamura et al., J Neurochem, 1991). We have obtained this monoclonal antibody from Dr. Kenji Tanaka previously in Okazaki and confirmed that it works well in chicken tissues. However, since this clone recognizes both PLP and DM-20 isoforms, it labels not only myelin-forming oligodendrocytes (MFOLs) but also newly formed oligodendrocytes (NFOLs) (Yokoyama et al., J Neurochem, 2025). Therefore, it is not ideal for determining whether nodal protein clusters are formed before myelin deposition.

Instead, we performed double immunostaining for MAG and AnkG between E12 and E15 to clarify the temporal relationship between myelin maturation and node formation. The results showed that detectable AnkG clusters along NM axons began to appear very sparsely around E13, coinciding with the emergence of MAG signals, and became more prominent with development. This temporal pattern does not match the definition of pre-nodal clusters, which are formed prior to myelination.

Although we cannot completely rule out the possibility of undetectable pre-nodal clusters or those composed of molecules other than AnkG, our results support the view that pre-nodal clusters are unlikely to play a major role in determining the regional difference in nodal spacing along NM axons. These new data have been added as Figure 2—figure supplement 1, and the relevant sections in the Results, Discussion, and Figure legend have been revised accordingly (page 5, line 4; page 10, line 7; page 29, line 1).